# Robustness of atmospheric trace gas retrievals obtained from low spectral resolution Fourier-transform infrared absorption spectra under variations of the interferogram length

Bavo Langerock[1], Martine De Mazière[1], Filip Desmet[1], Pauli Heikkinen[2], Rigel Kivi[2], Mahesh Kumar Sha[1], Corinne Vigouroux[1], Minqiang Zhou[3], Gopala Krishna Darbha[4], and Mohmmed Talib[4]

[1]Royal Belgian institute for Space Aeronomy, Ringlaan 3, 1180 Ukkel, Belgium
[2]Space and Earth Observation Centre, Finnish Meteorological Institute, Tähteläntie 62, 99600 Sodankylä, Finland
[3]Institute of Atmospheric Physics, Chinese Academy of Sciences, Beijing, China
[4]Indian Institute of Science Education and Research Kolkata, India

**Correspondence:** B. Langerock (bavo.langerock@aeronomie.be)

**Abstract.** This study examines the sensitivity of atmospheric trace gas column retrievals from ground-based Fourier-transform interferometer measurements to variations in the number of points in the recorded interferograms. Shortening an interferogram can be part of standard FTIR data processing and typically occurs with a convolution operation on the interferogram. Shortening will alter the leakage pattern in the associated spectrum and we demonstrate that the removal of a relatively small number of points from the interferogram edges creates a beat pattern in the difference of the associated spectra obtained from the original and shortened interferogram. For low spectral resolution interferometers the beat pattern may exceed the spectral noise level and if this occurs, the number of points in the interferogram can become a strong influence parameter for the retrieved trace gas. In a case study with formaldehyde retrievals obtained from low-resolution spectra in Sodankylä and Kolkata, we show that the retrieval software does not accurately model the leakage pattern and that interferogram shortening has a large effect on atmospheric gas column retrievals that exceeds the estimated retrieval uncertainty of approximately 15-30 %. This sensitivity of the retrieval algorithm to the length of the underlying low-resolution interferogram can be reduced by applying a non-trivial apodization such as Norton-Beer. For the Sodankylä case study the correlation between formaldehyde columns obtained from low- and high-resolution measurements increased from 0.72 (without apodization) to 0.93 (Norton-Beer strong apodized).

## 1 Introduction

Fourier-transform infrared (FTIR) interferometers have a long history in measuring atmospheric trace gas concentrations (De Mazière et al., 2018). The Network for the Detection of Atmospheric Composition Change (NDACC) uses interferometers with a high spectral resolution of approximately 0.005 cm$^{-1}$ in order to provide vertically resolved trace gas abundances. The Total Carbon Column Observing Network (TCCON) uses measurements with a spectral resolution of 0.02 cm$^{-1}$ to provide vertically integrated dry-air mole fractions (DMF) (Wunch et al., 2015). During recent years it has been demonstrated that FTIR interferometers with a low spectral resolution ranging from 0.2 cm$^{-1}$ to 0.5 cm$^{-1}$ have the ability to provide

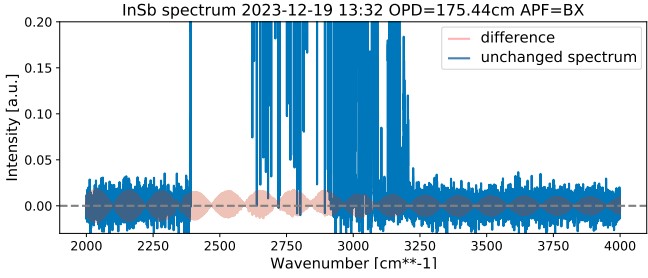 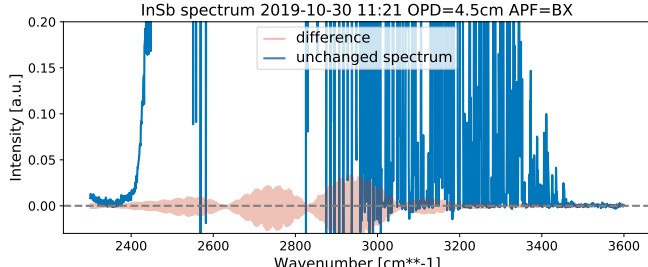

<table>
<tr><td>(a) Difference in spectra for a high-resolution example (0.005 cm$^{-1}$)</td><td>(b) Difference in spectra for a low-resolution example (0.2 cm$^{-1}$)</td></tr>
</table>

**Figure 1.** Difference in spectra for two boxcar (BX) apodized interferograms obtained after removing $\delta = 2^8$ points: (a) shows the beats in the difference where the length of the interferogram is $N \approx 5 \times 10^6$ points (obtained with a Bruker 125HR instrument), (b) shows the difference where the interferogram contains $N \approx 2 \times 10^5$ points (obtained with a Bruker Vertex70 instrument). Both spectra are recorded with an InSb detector. The header indicates the apodization function (APF) and the maximal optical path difference (OPD).

high-quality DMF data (Frey et al., 2019; Sha et al., 2020). FTIR instruments record interferograms that are converted to spectra from which atmospheric trace gas concentration are retrieved. This paper discusses the sensitivity of this processing chain to a (small) change in the number of points in the recorded interferogram. This dependence on the interferogram size has important implications when applying retrieval strategies used in the NDACC and TCCON networks directly on low spec-
25 tral resolution measurements. Up to our knowledge, this influence parameter in the retrieval chain was not considered before. This study builds on standard FTIR data processing methods, including zero-filling, phase correction, ramp correction and apodization (Herres and Gronholz, 1984) and on the general retrieval scheme applied in the above mentioned networks where a gas concentration is typically derived from a list of carefully selected spectral microwindows inside the observed spectra. For completeness we mention here that the COllaborative Carbon Column Observing Network (COCCON, Frey et al. (2019)),
dedicated to DMF retrievals from low spectral resolution measurements, applies by default a Norton-Beer apodization on the interferograms which reduces the dependence of the retrievals on the interferogram size (how this works will be explained in Section 3). NDACC and TCCON retrievals strategies typically use a boxcar apodization.

The operation that removes points from the tails of a recorded interferogram typically occurs when a convolution is performed on the interferogram prior to applying the discrete Fourier transform (DFT). For example, a standard practice consists
of resampling the interferogram to reduce its asymmetry and this operation uses a finite convolution with the sinc kernel (Forman et al., 1966, Eq. 13). The interferogram is typically shortened with half the size of the sinc kernel. Digital filtering uses a convolution in the time (or spatial[1]) domain (Herres and Gronholz, 1984, part 3). Another example is the Forman phase correction method which also applies a convolution with a non-standard kernel to reduce the phase in the spectrum (Forman et al., 1966). Figure 1 shows the effect caused by shortening interferograms that are obtained from two different instruments:
Fig. 1(a) contains a spectrum of an instrument measuring at high spectral resolution while Fig. 1(b) shows one at lower resolution (details on the resolutions and path differences are mentioned in the figure panels). Figure 1 reveals the appearance of

---

[1]time difference corresponds to the path difference divided by the speed of light

beats in the difference of the unchanged spectrum and the spectrum obtained after removing a few hundred points from the interferogram tails. For the low-resolution spectrum in Fig. 1(b), the amplitude of the beats is much larger than the noise level in the spectrum in the out of band region. Note that spectral differences can be taken point-wise as the number of points in the spectra is left unchanged even if the interferogram is shortened. This is due to the zero-filling operation which pads the interferogram with zeros such that the length of the signal becomes a power of two (Herres and Gronholz, 1984). For relatively small shortenings and in non-exceptional circumstances zero-filling therefore ensures that the number of points in the spectral domain is fixed.

Section 2 contains a short discussion to quantify the size of the beats using simulated spectra. Section 3 demonstrates how apodization can make the retrieval process more robust under such spectral leakage pattern perturbations. The final Section 4 studies the effect on formaldehyde retrievals, explains why the retrieval process does not model the leakage pattern accurately and demonstrates that the use of an appropriate apodization for low-resolution spectra stabilizes the atmospheric gas retrievals.

## 2 The Dirichlet kernel explains the beat pattern

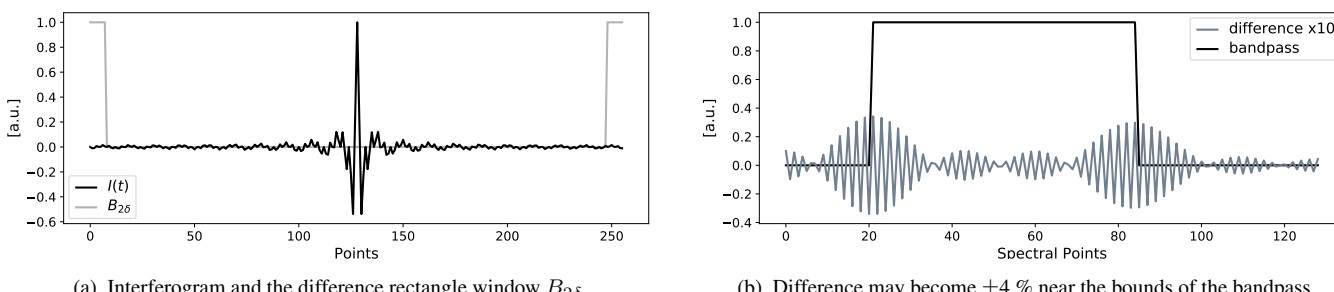

(a) Interferogram and the difference rectangle window $B_{2\delta}$

(b) Difference may become $\pm 4$ % near the bounds of the bandpass

**Figure 2.** Simulation example where the shortening operation is applied to a double sided interferogram shown in (a) that corresponds to an idealized bandpass model shown in (b) (only positive frequencies are shown in the spectral domain, interferogram size $N = 2^8$, bandpass width $L = N/4$ ). The effect in the spectral domain of removing $\delta = 2^3$ points at each end of the interferogram is shown. No apodization is used.

Figure 2 shows the effect of edge removal in a theoretical example using a symmetrical double sided interferogram $I[t], 0 \leq t \leq N$ that corresponds to an ideal bandpass model with width $L$. The right panel (b) shows the difference between the spectrum that corresponds to the original interferogram and the one where $\delta << N$ points are removed from both edges prior to the DFT. The standard processing chain that we adhere to applies zero-padding of the interferogram to the next higher power of 2 ($N = 2^8$ in this example) and thus the removal of a few points from the edge of the interferogram does not change the number of points in the spectral domain. The difference between both spectra, when considered in the time domain, corresponds to the portion of the interferogram that is removed. The difference in both spectra therefore equals the DFT of the product of the interferogram and an appropriately positioned rectangle window of length $2\delta$ as depicted in Fig. 2(a): DFT$(I[t]B_{2\delta}[t])$. The

DFT of a rectangle with length $\ell$ is related to the Dirichlet kernel (McClellan et al., 2016):

$$D_\ell(x) = \frac{\sin(\ell x/2)}{\sin(x/2)}, \text{ sampled at } x = \frac{2\pi}{N}k \text{ with } k \in \mathbb{N}.$$

From the convolution theorem, we can state that the difference of the two spectra is therefore a circular convolution of the bandpass with an appropriately time shifted Dirichlet kernel $D_{2\delta}(x)e^{i(N-1)x/2}$.

As shown in Fig. 2(b), this convolution creates a beat pattern in the difference. In this example the removal of $2\delta = 16$ points leads to an approximate 4 % difference in the spectra. The amplitude of the beats is related to the energy in the difference of both spectra, which in turn is related to the energy in the portion of the interferogram that is removed, being $I[t]B_{2\delta}[t]$ (Parseval's theorem). The beat amplitude therefore depends on the number of points removed $\delta$ and on the distance to the center burst since the energy in the interferogram signal is inversely proportional to the distance to the center burst.

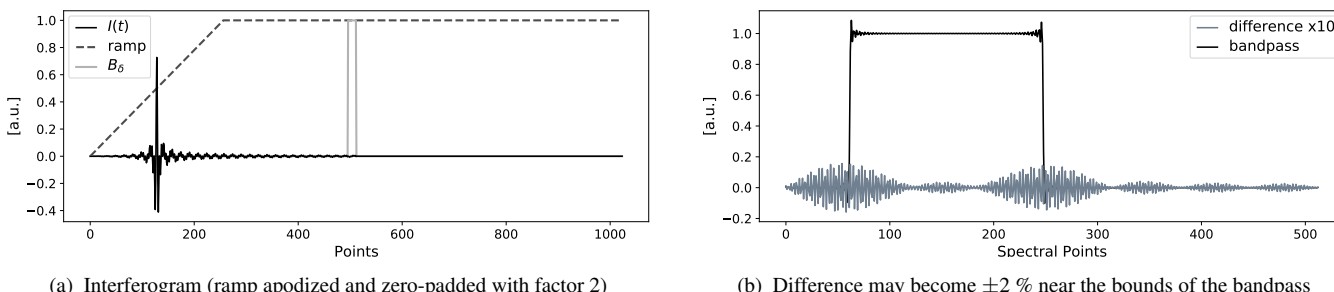

(a) Interferogram (ramp apodized and zero-padded with factor 2)     (b) Difference may become $\pm 2$ % near the bounds of the bandpass

**Figure 3.** Simulation example where the shortening operation is applied to a single-sided interferogram shown in (a) that corresponds to an idealized bandpass model shown in (b) (only positive frequencies are shown in the spectral domain, interferogram size $N = 2^9$, bandpass width $L = N/8$). The effect in the spectral domain of removing $\delta = 2^4$ points at the ends of the long arm of the single-sided interferogram is shown. The example uses a zero-filling factor of 2 and no additional apodization besides ramp is used.

For single-sided interferograms the ramp correction will ensure that the removal of $\delta$ points on the left shorter arm of the interferogram has a negligible effect, see Fig. 3. In this case only the right side rectangle window $B_\delta$ should be taken into account and the associated Dirichlet kernel takes a slightly different form due to a different time shift: $D_\delta(x)e^{-i(N_r-(\delta+1)/2)x}$ where $N_r$ is the number of points in the longer right arm.

## 3 Apodization

The appearance of beats is related to a perturbation in the leakage pattern underlying the DFT. Including apodization in the DFT will therefore reduce its sensitivity to such relatively small changes in the number of points. We consider the standard three Norton-Beer apodization windows from (Norton and Beer, 1976, 1977) although a more arbitrary strength could be used (Ntokas et al., 2023). Figure 4 shows that the shape of the beats remains when applying different apodizations, however they are reduced in amplitude compared to the situation without apodization in Fig. 3. For stronger apodizations, the energy in the window $B_\delta$ containing the removed points is lowered and so is the amplitude of the beats.

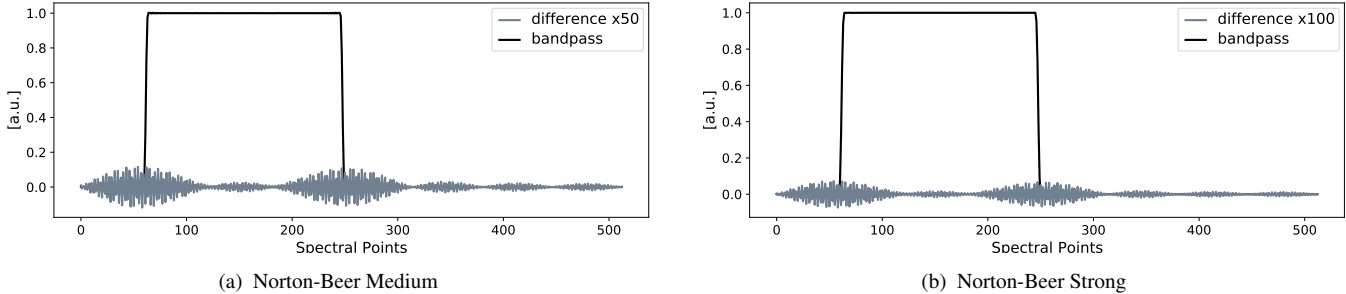

(a) Norton-Beer Medium

(b) Norton-Beer Strong

**Figure 4.** Similar as in Fig. 3, but here the interferogram is apodized with (a) the standard Norton-Beer medium window and (b) the Norton-Beer strong window. Apodizing reduces the beat amplitude with a factor of approximately 5 in (a) and 10 in (b) compared to Fig. 3.

Increasing the strength of the apodization will therefore reduce the dependence of a spectrum on changes in the number of interferogram points and can bring this dependence to the level of the noise in the out-of-band region. Figure 5 shows that for the low-resolution spectrum in Fig. 1(b), the stronger Norton-Beer apodization reduces the beats amplitude to the noise level.

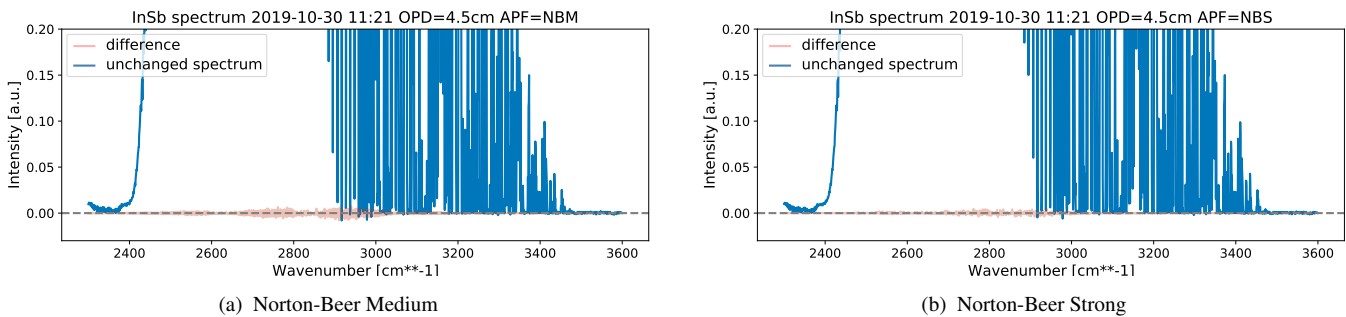

(a) Norton-Beer Medium

(b) Norton-Beer Strong

**Figure 5.** Similar to Fig. 1(b), but here the interferogram is apodized with (a) the standard Norton-Beer medium (NBM) window and (b) the Norton-Beer strong (NBS) window.

## 4 Case study: formaldehyde retrievals from low-resolution solar absorption spectra

For this case study we first consider a single day of measurements obtained from a low-resolution Bruker Vertex70 instrument (maximum optical path difference OPD equals 4.5 cm) deployed at the remote site Sodankylä (Finland, high latitude, Kivi and Heikkinen (2016)) during the 2019 campaign in the framework of the FRM4GHG project (Sha et al., 2020) and one day when the instrument was deployed at a polluted site in Kolkata (India, tropics). The purpose of this study is to check the robustness of the formaldehyde retrieval to small changes in the number of points in the interferogram. For that purpose formaldehyde columns are retrieved from 8 different sets of spectra obtained from the same underlying interferograms but with 4 different types of apodization applied (boxcar and 3 standard Norton-Beer) and with each interferogram shortened ($\delta = 2^8$) or left unchanged ($\delta = 0$). All retrievals follow the retrieval strategy that is currently adopted in the NDACC infrared

working group (Vigouroux et al., 2018). This strategy is applied to the low-resolution measurements without further change

because it was found that this gives the lowest bias when comparing to high-resolution measurements at Sodankylä (Sha et al., 2024). The strategy uses microwindows between 2760 cm$^{-1}$ and 2785 cm$^{-1}$ wavenumbers and makes the number of spectral points in the modelled spectrum much smaller compared to the measured spectrum. The estimated systematic uncertainty on the formaldehyde columns for this strategy applied to low-resolution spectra is of the order of 15-30 % (Vigouroux et al., 2018; Sha et al., 2024) depending on the pollution level. The retrievals are performed with the retrieval software package

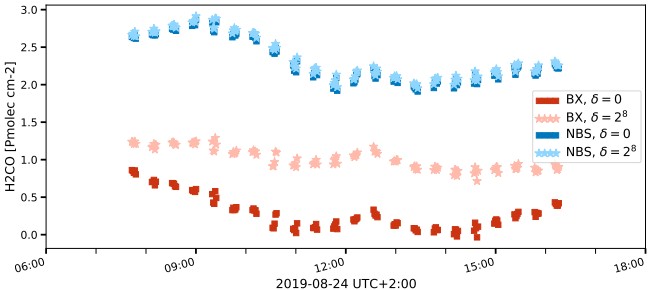

(a) Vertex70 (BX and NBS) formaldehyde columns at Sodankylä, shortened versus unchanged interferograms

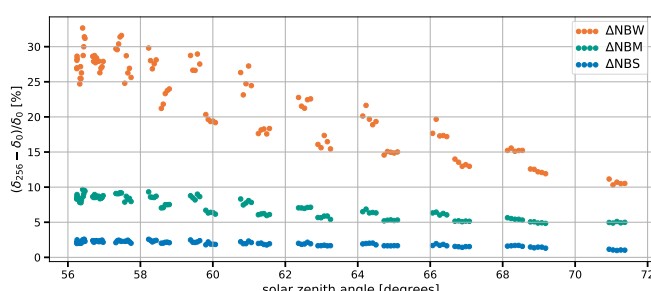

(b) Relative column differences between shortened and original interferograms for the three Norton-Beer apodizations at Sodankylä

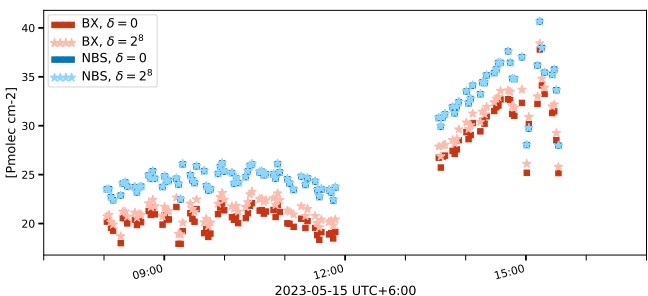

(c) Vertex70 (BX and NBS) formaldehyde columns at Kolkata, shortened versus unchanged interferograms

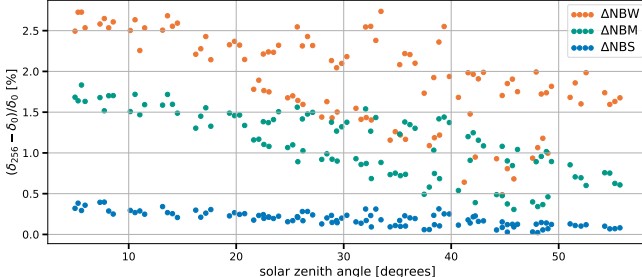

(d) Relative column differences between shortened and original interferograms for the three Norton-Beer apodizations at Kolkata

**Figure 6.** Sensitivity study in the retrieval of formaldehyde during one day, where the formaldehyde is retrieved from spectra obtained with either a boxcar (BX, red) or Norton-Beer strong (NBS, blue), medium (NBM, green) and weak (NBW, orange) apodization and where either no points ($\delta = 0$) are removed or $\delta = 2^8$ points are removed.

SFIT2 (Pougatchev et al., 1995) which was updated to SFIT4 (Hannigan et al., 2024) and is one of the standard retrieval algorithm implementations used in the NDACC infrared working group. The forward model component in SFIT4 takes into account the size of the interferogram via the maximal OPD to simulate the leakage in the modelled spectrum. This is done by a truncation in the time domain after applying the DFT on the modelled spectrum. The truncation index is obtained from the measured maximal OPD using the direct proportion between the OPD and the number of points in the interferogram: more

precisely, it is calculated by multiplying the number of points in the modelled interferogram with the ratio of the measurement maximal OPD and the modelled OPD. This index is typically invariant under small perturbations of the measurement OPD

value because the scaled truncation index requires a rounding to an integer. In the specific case of the Vertex70 spectra for Sodankylä, we have compared two sets of retrievals with maximal OPD equal to 4.5 cm (no modelled shortening of the IFG) and maximal OPD 4.49 cm (modelled shortening of the IFG with $\delta = 2^8$ points out of 171130). The difference in retrieved formaldehyde columns for the Sodankylä measurements used in Fig. 6 is less than 2 % and is due to a dependence in the forward model wavenumber spacing on the maximal OPD. This difference can be considered negligible compared to the retrieval uncertainty. In the following all retrievals were done with a fixed maximal OPD value.

Panels (a) and (c) in Fig. 6 shows that the boxcar (BX) retrievals are strongly dependent on the number of points in the interferogram while the NBS apodized retrievals are (nearly) invariant. The formaldehyde retrieval windows (ranging between 2760 cm$^{-1}$ and 2785 cm$^{-1}$) coincide with a beat from the pattern in Fig. 1(b) which partly explains this strong dependence.

Panels (b) and (d) in Fig. 6 show that the column difference (shortened - unchanged) depends on the chosen strength of the apodization: for the clean site Sodankylä, the relative differences range from values up to 30 % for NBW to 10 % for NBM and below 5 % for NBS. The Sodankylä retrieved columns from the NBS apodized spectra have a strongly reduced sensitivity to the number of points which becomes negligible when compared to the retrieval uncertainty budget. For the polluted site at Kolkata, depicted in (c) and (d), the relative differences are much reduced but reveal the same pattern as for Sodankylä. Note

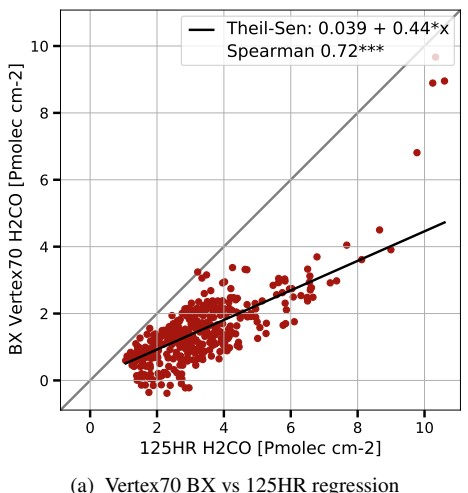
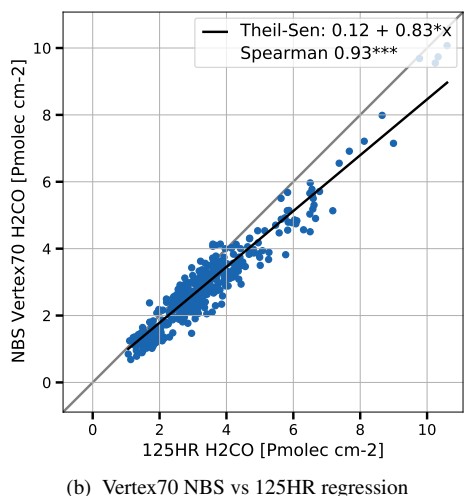

(a) Vertex70 BX vs 125HR regression

(b) Vertex70 NBS vs 125HR regression

**Figure 7.** Correlations for formaldehyde columns using (a) boxcar and (b) Norton-Beer strong apodizations for the Vertex70 Sodankylä campaign data in 2019 compared to the NDACC 125HR observations, using coinciding measurements in 15 minutes averaged data.

also that Fig. 6 shows that both the perturbation in the number of points and the use of different apodizations propagate to different airmass dependence effects in the differences of the formaldehyde columns.

For Sodankylä the NBS apodization reduces the sensitivity on the interferogram size well below the reported uncertainty and is therefore chosen for a comparison between the Sodankylä Vertex70 retrievals and the Sodankylä Bruker 125HR retrieved columns that are publicly available from the NDACC rapid delivery database (Kivi et al., 2025). To quantify the bias and correlation with the NDACC 125HR data we have retrieved all Vertex70 FRM4GHG 2019 campaign spectra using a BX and

a NBS apodization. Figure 7 uses a fifteen minute average of the NDACC retrievals and Vertex70 retrievals and only fifteen minute intervals that coincide are shown (Sha et al., 2024). The overall bias of low-resolution vs high-resolution (NDACC) retrieved columns reduces from approximately -55 % for the low-resolution columns retrieved from the BX apodized spectra to -15 % for the low-resolution retrievals from the NBS apodized spectra. Similarly the correlation coefficient (Spearman, p-value < 0.01) between the low-resolution and high-resolution retrievals increases from 0.72 for the low-resolution BX retrievals to 0.93 for the low-resolution NBS retrievals.

The effect of apodization in the retrieval results does not contradict the conclusions from Amato et al. (1998), where it is shown that retrieval results are invariant under different apodizations if one takes into account that the measurement error in the spectral domain is no longer random noise after applying apodization. The retrieval software SFIT, typically used with high-resolution spectra, always considers the measurement error as random noise and therefore produces different retrieval results after applying apodization. As mentioned in Amato et al. (1998), apodizing an interferogram reduces the higher spectral resolution information in the spectrum. For the case study considered in this paper, the formaldehyde columns obtained from the apodized low-resolution column agree better with the high-resolution compared to the non-apodized boxcar data which shows that the information loss in the spectra is negligible compared to the non-accurate modelling of the leakage pattern in the retrieval scheme.

For completeness, it should be mentioned that the high-resolution NDACC retrievals are made from BX apodized spectra. A robustness test as in Fig. 6, showed no significant dependence of the 125HR retrieved columns on the interferogram size nor on the chosen apodization: all relative differences for the Sodankylä 125HR are below 0.1 %.

## 5 Conclusions

We have demonstrated that a shortening of an interferogram creates a perturbation in the leakage pattern that resembles a beat pattern in the difference of the associated spectra. The amplitude of the beats is related to the energy of the signal in the truncated tail of the interferogram and is therefore larger for low-resolution interferograms. We have shown examples using idealized bandpass models and real measurements where the beats exceed the noise level in the out-of-band region. Retrieval software packages may not be accurate enough to model such small changes in the interferogram size due to rounding errors in the calculation of the leakage pattern. This was demonstrated with the retrieval of atmospheric formaldehyde columns from low-resolution measurements which turned out to be unstable if no appropriate apodization is used. In this case study we have chosen the strength of the apodization (e.g. Norton-Beer weak/medium/strong) such that the retrieved formaldehyde columns become stable under shortenings of the underlying interferograms in the sense that retrieved column differences are within the estimated formaldehyde uncertainty budgets. The bias between the low- and high-resolution data decreased from -55 % if no apodization is applied to the low-resolution interferograms to -15 % if a NBS apodization is used. The correlation increased from 0.72 (no apodization) to 0.93 (NBS). The retrieved formaldehyde columns obtained from the high-resolution NDACC measurements are not significantly affected when changing the apodization.

There are many parameters that may influence the sensitivity to the spectrum leakage pattern of the retrieved trace gas product: the signal to noise ratio, the size of the microwindows, the concentration level of the retrieved gas, the spectral resolution of the measurements (the maximal optical path difference). We recommend that the interferogram length should be considered as an important influence parameter in the development of a trace gas retrieval strategy for measurements from a lower resolution FTIR interferometer. Different apodizations should be considered in stability tests on the retrieved trace gas products from shortened interferograms. We recommend that the selection of the final apodization strength is done such that the sensitivity to shortening becomes negligible compared to the typical retrieval uncertainty caused by the measurement noise and forward model uncertainties (such as spectroscopy, solar zenith angle, ...), see Rodgers (2000).

*Code and data availability.* SFIT4 (Hannigan et al., 2024) is a radiative transfer and atmospheric constituent profile retrieval algorithm for use with spectra recorded in solar absorption and emission across the infrared spectrum. The Sodankylä 125HR formaldehyde data (Kivi et al., 2025) used in this publication were obtained as part of the Network for the Detection of Atmospheric Composition Change (NDACC rapid delivery archive, https://ndacc.org, last public access: 13 February 2025). The other data used in this publication can be provided upon request to the corresponding author.

*Author contributions.* BL wrote the paper, did the analysis and sensitivity studies. CV implemented the NDACC formaldehyde strategy for the Vertex70 low-resolution spectra and the presentation of the statistics in the 2019 Sodankylä time series. MDM, MKS, RK and PH contributed with the Vertex70 spectra from the FRM4GHG campaign in Sodankylä, in addition RK and PH with the 125HR spectra from Sodankylä and MDM, MKS with the spectra from Kolkata. GKD and MT for the maintenance and support with the instrument operations at Kolkata. MKS, FDS and MZ contributed with valuable suggestions to understand the cause of the beat patterns. All authors have read and agree to the published version of the manuscript.

*Competing interests.* The authors declare that they have no conflict of interest.

*Acknowledgements.* The authors thank Philippe Demoulin for useful discussion on the implementation details of the DFT at the Jungfraujoch station. The BELSPO ProDEx project TROVA-E2 (PEA 4000116692), SVANTE (4000132151/20/NL/FF/ab), FRM4GHG (4000117640/16/I-LG and 4000136108/21/I-DT-lr), and ACTRIS-BE projects provided funding for the observations used in this work. Nicolas Kumps, Christian Hermans and Christof Petri for the operation and set up of the instrument during the Sodankylä campaign.

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
