# Peer review of "Robustness of atmospheric trace gas retrievals obtained from low spectral resolution Fourier-transform infrared absorption spectra"

_EGUsphere, 2024_

## Author Response (AR2)

**Reply on Report #1**

We experienced the report of the anonymous referee and the qualification "major revision" as unexpected. We suspect that there are misunderstandings in our previous replies to the referee.

A possible important misunderstanding might be related to the fact that the actual number of points in an interferogram recording is determined in an approximative way and is not "absolute". Approximative in the sense that the user operating the instrument requests a certain spectral resolution in cm-1 which is then translated into a maximal optical path difference (in cm) and subsequently into a number of interferogram points (OPD ~ number of interferogram points multiplied with half the instruments' laser wavelength being of the order 3e-5cm [Herres 1984, Part 2 §1.1]). A few hundred points more or less in the interferogram will not affect the requested spectral resolution in cm**-1 units. Standard operations on interferograms (convolution/resampling operations) will shorten the interferogram, but these operations are typically considered harmless as the requested spectral resolution is not affected.

In our understanding shortening an interferogram is a standard practice.

The main finding of the paper is: the retrieval of atmospheric trace gas from low spectral resolution FTIR measurement may be strongly influenced by the number of points in the underlying interferogram. Up to our knowledge this was never documented before.

We have chosen to write this finding in short paper by means of a case study: when shortening the interferograms from the Vertex70 Sodankyla instrument by 256 points, the associated formaldehyde columns increase significantly, i.e. more than the reported formaldehyde uncertainty.

We would like to emphasize that the "methodology" and "clarifying context" of the paper is straightforward and simple: shorten the number of points in the recorded interferogram, apply the standard retrieval algorithms and the outcome as a trace gas column is "significantly" different.

We have chosen not to include technical details on FTIR data processing (phase correction methods, dc-correction, etc), because these details do not contribute to the understanding of the observed sensitivity behavior. Nor do the details of the retrieval algorithms used in the FTIR networks NDACC/COCCON/TCCON. It is our belief that these details would overload the paper and mask the communication of the main finding.

On the other hand, the paper does explain that the observed sensitivity to shortening is caused by a perturbed leakage pattern in the spectrum which may become larger than the spectral noise for low spectral resolution measurements combined with the retrieval software being unable to correctly model the leakage pattern. We have chosen to include only those technical details that are relevant for understanding the observed sensitivity.

It seems however that what we thought was a good approach to communicate our findings, a short case study paper, is interpreted completely differently by the referee. Because we are convinced a misunderstanding is playing here, we did not rewrite the manuscript and choose to resubmit a minor revision. We hope the referee understands this.

*First, the title is too broad and could be refined to better highlight the focus on interferogram analysis, something like this?: "Robustness of atmospheric trace gas retrievals through interferogram analysis of low-resolution FTIR spectra"*

> We agree that the title is broad in the sense that many parameters can be perturbed to test the "robustness" of a trace gas retrieval. We have chosen to make the title broad and to specify in the first sentence of the abstract the detail of what is actually being perturbed. The referee suggests to use "interferogram analysis" in the title. It is our feeling that "interferogram analysis" remains a broad concept and we are not convinced that this would help the reader to understand in more detail what is perturbed directly from the proposed title.

We have now changed the title to "Robustness of atmospheric trace gas retrievals obtained from low spectral resolution Fourier-transform infrared absorption spectra under variations of the interferogram length".

*The tone and style of the paragraph below is not appropriate, it is dismissive. Peer-reviewed articles should aim for accessibility to a broader audience, including those who may not specialize in FTIR data processing. "We assume that the reader is familiar with the standard operations in FTIR data processing such as zero-filling, phase correction, ramp correction and apodization (Herres and Gronholz, 1984) and with …"*

We regret that the referee finds the tone in his paragraph dismissive. It was never our intention to write in a dismissive tone or to belittle any reader. From the suggested change, we understand the referee is displeased with the phrasing "the reader is familiar". We thought this phrasing is commonly used (see scholar.google.com with +66700 hits from various publications in different subjects). Because we are non-native English speakers, we will rely on the judgement of the referee and follow the suggestion in the revised version.

We agree that "review" articles should aim for a broader audience, but disagree that this should be the case for all peer-reviewed articles (many examples in existing literature can serve as counter examples). This paper is not a review paper: the purpose is to show by means of a case study that the retrieval processing chain may be sensitive to shortening of the interferogram.

*The authors claim to have included motivation for their work in page 1 line 19. However, I did not find this, the introduction is brief and does not sufficiently address why this analysis is necessary or how it contributes to the field. I recommend expanding the introduction to clearly state the problem, its broader implications for FTIR data processing, and why interferogram shortening is significant/relevant in current practices. Is shortening an interferogram a very common approach? Are there recent references/studies stating this approach?*

We assume a misunderstanding occurred. Page 1 line 19 refers to the initial submission, maybe that is why the referee did not find the motivation. In our first reply to the referee we explicitly mentioned that (valid) convolution operations shorten the interferogram and that examples are part of the introduction. To answer the referees question: yes, shortening, although unintentional and a side effect of convolution operations, is part of standard FTIR data processing. This statement is now part of the abstract.

The referee insists on explaining "why we consider shortening" of the interferogram. We believe it is not relevant: it is a standard practice because the actual length of an interferogram is an approximative number for the requested spectral resolution (see above), the length of an interferogram is not absolute and changes due to convolution operations are considered harmless.

The case study in the paper shows that the length of the interferogram may be an important influence parameter in the retrieval of atmospheric trace gases. As far as we know this influence parameter was never described/studied in a publication. We hope that these considerations, now included in the abstract, are sufficient to convince the referee.

*The dismissal of suggestions to include more detailed explanations of FTIR practices as "out of scope" is concerning. Understanding how high- and low-resolution instruments handle data processing, including details about software and routines (e.g., zero-filling, phase correction, apodization), is critical for contextualizing the findings. This is especially true if they claim "Robustness of atmospheric trace gases".*

This comment relates to a previous comment: this paper is not a review paper and we assume the reader is familiar with standard FTIR processing. There is no essential difference in handling measurements from high or low-resolution instruments. The networks NDACC/TCCON/COCCON can make different choices in how the standard FTIR

processing steps are applied (eg which phase correction or apodization). These differences are not relevant to understand the sensitivity to the number of points.

We do not understand the last sentence in this comment: we never claimed that the paper gives a complete discussion on all possible influence parameters. The paper is about "robustness of atmospheric trace gases under variations in the interferogram length" .

*Motivation should connect the case study to existing gaps in knowledge and potential applications for FTIR observations. Although the authors made some adjustments to the abstract, it remains brief and lacks a strong motivating statement. I suggest explicitly stating the significance of interferogram shortening in current practices and including quantitative findings (e.g., HCHO results).*

We added quantitative findings from the case in the abstract and added a sentence in the introduction that the length of the interferogram was not considered before as an influence parameter and that it may have an impact when applying retrieval strategies for high-resolution measurements to low-resolution measurements.

*The authors reject (or missed to respond) the suggestion to assess results using control-based cell spectra, but I believe this would strengthen the study. Cell spectra provide a controlled environment that could validate the findings and test the impact of instrument line shape on retrievals. Without this, it is difficult to assess the robustness of the conclusions. This is further linked to the diurnal variation of Fig 6, where authors suggest that diurnal variation in SNR is the likely cause of the diurnal variation in Fig 6 but do not include a quantitative analysis. Cell spectra will further help in this.*

The title states that the paper is about atmospheric trace gas retrievals. Cell spectra retrievals are not. Cell spectra are treated differently: prior to retrieving the instrument line shape, the background is removed from the cell spectra and the (linefit) algorithm focuses on the sharp absorption lines from the gas cell. The robustness of the retrieval of cell spectra is therefore considered out of scope.

If the referee refers to the influence of the instrument line shape on these formaldehyde columns, then this is taken into account in this particular formaldehyde retrieval by co-retrieving the ILS (Vigouroux 2018). The typical effective apodisation for these instruments is good (2%-5%, Frey 2019, Sha 2019, Alberti 2022). The ILS apodisation is thus of a completely different order compared to the Norton-Beer apodisations required in this case study to make the retrieval robust.

**List of changes**

- title
- the example of shortening in a convolution is now part of the abstract
- abstract was adapted and includes a specification of "large effect on atmospheric gas column retrievals": the effect of shortening on the formaldehyde column can be larger than the reported measurement uncertainty
- in the introduction we added a sentence "This dependence on the interferogram size has important implications for NDACC and TCCON retrieval algorithms when applied on low spectral resolution measurements. Up to our knowledge, this influence parameter in the retrieval chain was not considered before." This highlights possible implications of the main findings in the paper.
- We added a DOI to the NDACC Sodankyla formaldehyde NDACC data set and included it in the references.